# Object-level Semantic and Spatial Distillation for Open Vocabulary Detection

**Zitong Li** [1]   **Jinzhuo Wu** [1]   **Fukang Zhao** [1]   **Xinyue Wang** [1]   **Jun Chen** [2]   **Zhuo Cheng** [1]   **Dapeng Luo**[✉ 1]

## Abstract

Recent Open-vocabulary Object Detection (OVD) approaches adapt CLIP through region-level distillation to improve semantic alignment for novel categories. However, the distilled regional features are often used for both classification and localization, enhancing semantic consistency at the expense of spatial fidelity. To resolve this, we propose Object-level Semantic and Spatial Distillation (OSSD), a two-stage framework that explicitly decouples semantic and spatial feature learning. OSSD first distills object-level semantics from CLIP's global [CLS] embeddings to enhance region discrimination, and then injects fine-grained spatial and structural priors via spatial distillation from a detector trained only on COCO base categories. Furthermore, we propose a Location Quality Estimation Head (LQEH) that predicts class-agnostic localization quality, complementing objectness confidence to improve the novel-object perception. Extensive experiments show that our method achieves 49.2 AP50 on the OV-COCO benchmark. exceeding the best previous result by 3.6%, On the OV-LVIS benchmark, our method reaches 40.5 mAP on novel categories, outperforming previous state-of-the-art methods.

## 1. Introduction

Open-vocabulary object detection (OVD) (Zareian et al., 2021) aims to recognize and localize visual concepts beyond predefined training categories, presenting a significant challenge for traditional closed-set detectors. The emergence of Vision-Language Models (VLMs) (Li et al., 2023; Radford et al., 2021; Sun et al., 2023) has opened new opportunities for addressing this challenge. Trained on large-scale

[1]School of Mechanical Engineering and Electronic Information, China University of Geosciences (Wuhan), Wuhan 430074, China [2]School of Artificial Intelligence and Automation, China University of Geosciences (Wuhan), Wuhan 430074, China. Correspondence to: Dapeng Luo <luodapeng@cug.edu.cn>.

*Proceedings of the 43rd International Conference on Machine Learning*, Seoul, South Korea. PMLR 306, 2026. Copyright 2026 by the author(s).

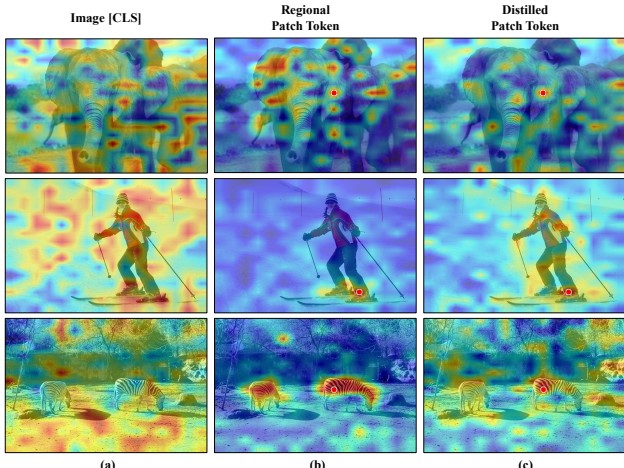

*Figure 1.* Visualization of token-level similarity maps from different feature representations. All maps are obtained by computing the cosine similarity between a reference token (highlighted in red.) and all other visual tokens. (a) Using the CLIP image-level [CLS] token as the reference. (b) Using regional patch tokens as the reference. (c) After self-distillation, part of their spatial distinctiveness is lost.

image-text pairs, VLMs learn aligned visual-semantic representations that demonstrate remarkable zero-shot recognition capabilities, providing a promising foundation for open-vocabulary understanding.

Building on this foundation, recent OVD methods leverage VLMs by equipping detectors with text encoders and learning end-to-end region–text alignment. These approaches can be categorized into two main directions. Localization-driven methods focus on enhancing region proposal generation to discover more potential novel category bounding boxes through robust localization-aware loss (Jeong et al., 2024), adaptive region selection (Zheng & Liu, 2024), and pseudo-label generation (Wang et al., 2024; 2025). However, these methods face an inherent limitation: VLMs are pre-trained on global image-text pairs, while object detection requires fine-grained region-level recognition. This semantic-spatial mismatch (Wu et al., 2023c) limits the effectiveness of directly applying VLM features to local object classification, motivating the need for better alignment strategies.

The second direction, semantic-driven methods, addresses this limitation by improving vision-language model (VLM)

feature adaptation for region classification in the detection task. These approaches employ techniques like region-specific prompts (Wu et al., 2023c; Li et al., 2024) or self-distillation (Naeem et al., 2024; Wu et al., 2023b; 2024) to bridge the gap between VLMs' global representations and the local recognition requirements of detection. In particular, distillation-based methods refine CLIP by aligning region-level representations pooled from the global feature map with CLIP's global [CLS] embeddings of the corresponding image crops, thereby improving region classification performance in open-vocabulary detection (OVD). However, in most existing approaches, the distilled regional features are simultaneously used for both object localization and classification, which often suppresses fine-grained spatial and structural details that are critical for accurate localization. This leads to degraded detection performance and reveals a fundamental trade-off between recognition and localization in OVD.

To further elaborate on the above observation, we conduct an in-depth analysis of CLIP's visual attention maps. As shown in Figure 1(a), CLIP's [CLS] token tends to capture global information including background regions and contextual cues rather than fine-grained object details (Li et al., 2022). In contrast, regional patch tokens (Figure. 1(b)) inherently preserve richer spatial and structural information, which is crucial for precise localization. However, when the regional feature embeddings are distilled to mimic CLIP's image-level [CLS] representation of the corresponding image crops (Figure. 1(c)), they inherit this background-biased characteristic. Such a shift can be detrimental to downstream localization tasks, where spatially discriminative and fine-grained features are essential for accurate object grounding and detection.

Moreover, existing OVD methods suffer from an inherent limitation: detectors trained only on base categories tend to assign higher confidence scores to proposals containing seen objects, while misclassifying novel category instances as background. This occurs because most OVD approaches solely rely on objectness confidence to determine whether a region corresponds to a valid object. However, this confidence is learned exclusively from base-category annotations, causing the detector to associate high objectness scores primarily with seen objects while suppressing novel-category instances as background.

Based on the above analysis, we propose Object-level Semantic and Spatial Distillation (OSSD), a two-stage feature distillation framework that decouples and separately optimizes semantic and spatial feature learning. Our approach first distills object-level semantic features from CLIP's global [CLS] tokens to inject instance-level semantic cues, then performs spatial distillation using a detector pre-trained solely on the base categories of the COCO dataset (Lin et al.,

2014). A Semantic-to-Spatial (S2S) adapter is designed to isolate the features involved in semantic and spatial distillation. This allows the model to maintain high spatial fidelity for localization while still leveraging CLIP's rich semantic knowledge for both base and novel category recognition. Furthermore, to enhance spatial discrimination on novel categories, we propose a Location Quality Estimation Head (LQEH) that learns to regress the localization quality of each detection proposal. Localization quality is inherently class-agnostic, which can be trained only on base categories and naturally generalized to novel objects. By jointly considering both objectness confidence and spatial precision, our approach can effectively distinguish well-localized novel objects from low-quality background proposals, thereby mitigating base-class bias and improving detection performance on unseen categories.

The main contributions of this paper are summarized as follows: (i) We propose a two-stage feature distillation framework that effectively bridges the spatial-semantic gap and enhances localization ability without compromising CLIP's pre-trained alignment; (ii) We introduce a Location Quality Estimation Head (LQEH) that complements objectness confidence with explicit spatial quality assessment in a class-agnostic manner, thereby suppressing low-quality background proposals and alleviating base-class bias; (iii) OSSD consistently outperforms existing state-of-the-art methods on open-vocabulary detection (OVD) benchmarks. On the OV-COCO benchmark, our method improves the $AP_{50}^{Novel}$ of novel categories over the previous best method OV-DQUO (Wang et al., 2025) by 3.6 points. On the OV-LVIS benchmark, OSSD achieves 40.5 $mAP_r$ on rare categories, also outperforming the previous state-of-the-art method (Wang et al., 2025).

## 2. Related Works

### 2.1. Open-vocabulary Object Detection Methods

Open-vocabulary object detection (OVD) aims to detect and recognize objects from arbitrary categories, including unseen categories during training. Early work such as OVR-CNN (Zareian et al., 2021) first formulates OVD by aligning region features with textual nouns extracted from image captions, enabling detectors to generalize beyond fixed label spaces. Then, with the advent of large-scale vision–language models (VLMs) such as CLIP (Radford et al., 2021), recent methods increasingly exploit pre-trained image–text representations to enhance open-vocabulary recognition.

A representative line of research aims to improve object discovery for novel categories by enhancing the region proposal or localization stage. ProxyDet (Jeong et al., 2024) empirically shows that mixing base-class representations

can generate proxy-novel classes that approximate novel categories. AggDet (Zheng & Liu, 2024) proposes to aggregate confidence estimates by jointly modeling class-agnostic localization quality and text-guided prototype similarity. OV-DQUO (Wang et al., 2025) generates pseudo-labels for unknown objects recognized by an open-world detector, thereby alleviating the confidence bias between base and novel categories. However, the gap between global VLM representations and region-level detection hampers local object recognition, necessitating improved alignment strategies.

Another prominent direction aims to enhance region-level semantic representation by directly adapting or refining VLM features for detection. CORA (Wu et al., 2023c) leverages a lightweight region prompt to adapt frozen VLM features for more accurate region-level recognition. RegionCLIP (Zhong et al., 2022) fine-tunes CLIP on RPN-generated region crops (Ren et al., 2015) to alleviate the domain shift and enhance region-aware representations. CFM-ViT (Kim et al., 2023a) adapts CLIP during pretraining by augmenting the image–text contrastive objective with feature masking and positional embedding dropout. However, rather than implicitly inducing domain bias through pseudo-labeling or prompt-based adaptation, recent state-of-the-art methods typically adopt distillation-based strategies, which explicitly inject class-agnostic semantic and spatial priors while preserving CLIP's pretrained semantic space.

## 2.2. Distillation based OVD Methods

Early works primarily focus on distilling CLIP knowledge into object detectors to alleviate domain shift and improve compatibility with region-level representations. For example, ViLD (Gu et al., 2021) is the first to distill VLM classification knowledge into object detectors by aligning detector-generated region embeddings with corresponding VLM features, while BARON (Wu et al., 2023a) extends this by aligning bag-of-regions representations with global image embeddings extracted from VLMs. OADP (Wang et al., 2023) further introduces hierarchical distillation at object, block, and global levels, enabling multi-granularity semantic alignment between detection features and VLM representations.

More recent approaches (Naeem et al., 2024; Wu et al., 2023b; 2024) instead adopt self-distillation at the pretraining stage to enforce consistency between CLIP's regional patch features and image crop features. CLIPSelf (Wu et al., 2023b) pioneers a self-distillation framework that aligns dense visual features with corresponding image-level representations, thereby endowing CLIP with region-aware visual features. CLIM (Wu et al., 2024) further extends this idea by performing self-distillation on dense features through multi-image composition, encouraging CLIP to capture richer regional semantics.

Nevertheless, we identify that these methods reuse distilled region features for both localization and classification, where improved semantic alignment often comes at the expense of spatial precision, leading to a trade-off that limits detection performance.

## 2.3. Novel Object Perception Strategy

In open-vocabulary object detection, a key challenge lies in effectively perceiving and modeling novel objects during both training and inference. OW-DETR (Gupta et al., 2022) identifies potential unknown objects by exploiting activation patterns in feature maps, based on the observation that foreground regions typically elicit stronger responses than background regions. PROB (Zohar et al., 2023) addresses unknown object identification by modeling the distribution of output logits and decoupling background, known, and unknown predictions. Building on the insight that foreground regions exhibit higher variability while background regions change more monotonously, MEPU (Fang et al., 2025) introduces a Reconstruction Error-based Weibull (REW) model that assigns likelihood scores to region proposals potentially corresponding to unknown objects. While effective, these approaches leverage indirect indicators (e.g., activation intensity, logit distributions, reconstruction errors) shaped by base-class supervision, which may not generalize reliably to novel categories. Unlike these methods, our approach introduces a localization quality prediction head that is inherently class-agnostic and generalizes seamlessly to unseen objects, thereby enhancing novel object discrimination.

## 3. Method

### 3.1. Overview

In this section, we introduce Object-level Semantic and Spatial Distillation (OSSD), a two-stage distillation framework designed to decouple semantic and spatial feature learning and optimize them separately. Figure 2 provides an overview of OSSD. We begin by detailing the object-level semantic distillation stage, where CLIP's regional representations are enhanced using object proposals generated by a pretrained RPN. Next, we present a spatial distillation stage for distilling spatially enriched features from a pre-trained detector, with a Spatial-to-Semantic (S2S) adapter to isolate the feature representations of the two distillation stages. Finally, we introduce the Localization Quality Estimation Head (LQEH), which learns class-agnostic localization quality from base categories and improves the discrimination of well-localized novel objects from background proposals.

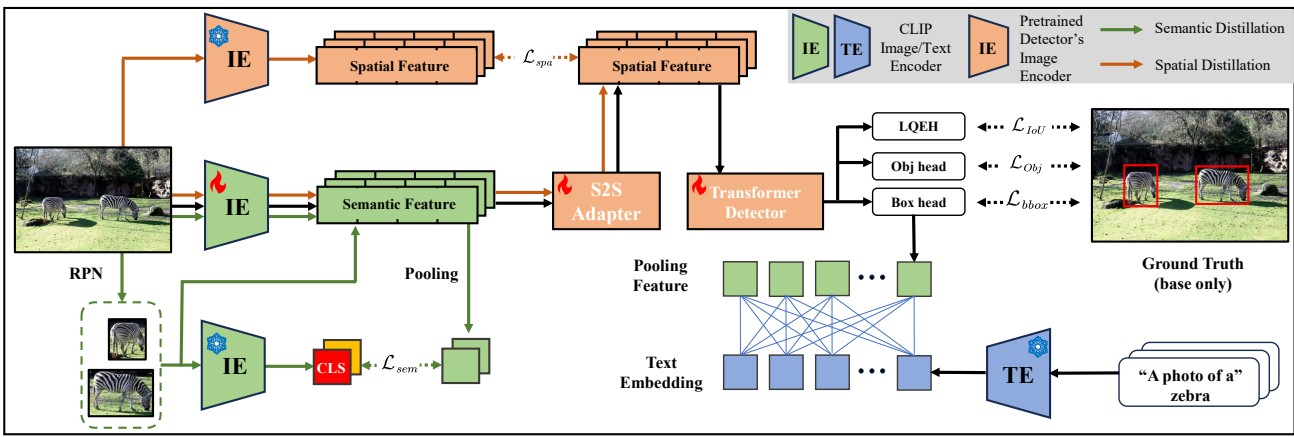

*Figure 2.* **Overview of OSSD. Semantic Distillation Pathway (green arrows):** Region proposals are encoded by a frozen CLIP teacher, and student region features are aligned with the teacher's image-level [CLS] representation to inject instance-level semantic cues. **Spatial Distillation Pathway (orange arrows):** A pre-trained detector's image encoder provides localization-aware supervision, transferring boundary-sensitive and structural features to the S2S adapter for improved bounding box regression. **LQEH (black arrows):** Operates on spatially enhanced features to regress localization quality, enabling more reliable ranking beyond objectness confidence.

### 3.2. Two-Stage Distillation Framework

**CLIP's Patch Feature Extraction.** In conventional open-vocabulary detectors, region proposals are typically classified by pooling CLIP's patch-level features into a single region representation. During CLIP pretraining, a global [CLS] token is explicitly optimized to align with the image-level text embedding via contrastive learning, while patch tokens are trained indirectly under the same global objective. Consequently, although region-level representations are formed by pooling patch tokens, these features are still biased toward holistic image semantics rather than instance-specific cues. To address this limitation, we extract localized patch-level features from CLIP's ViT encoder and perform semantic distillation to adapt them for region-aware recognition.

The CLIP ViT encoder consists of multiple stacked transformer attention layers (e.g., ViT-B/16 contains 12 attention layers). Let the input to the last residual attention layer be $X = \{x_0, x_1, ..., x_{N-1}\}$ with $x_i \in \mathbb{R}^{1 \times D}$, where $x_0$ denotes the [CLS] token and the remaining $h \times w$ tokens correspond to image patches ($N = 1 + h \times w$). In the last layer, the attention computation can be formulated as:

$$\text{Attn}_{qk}(X) = \text{Softmax}\left(\frac{QK^\top}{\sqrt{d_k}}\right)V, \qquad (1)$$

$$Z = \text{FFN}\left(\text{Attn}_{qk}(X) + X\right), \qquad (2)$$

where $\text{Attn}_{qk}$ denotes the self-attention weights, $Q, K, V$ denote projection layers and FFN is a feed-forward network. Normalization layers are omitted for clarity. After the last layer, the global semantic representation is captured by the [CLS] token $Z[0]$, while the remaining patch embeddings $Z[1 : h \times w]$ can be reshaped into a regional patch feature map $X_{\text{patch}} \in \mathbb{R}^{C \times H \times W}$.

However, the gap between CLIP's image-level pre-training and region-level OVD tasks leads to spatially inconsistent attention. Recent studies (Pu et al., 2023; Ye et al., 2023) mitigate this issue by modifying the attention formulation in the last CLIP attention layer to encourage more spatially coherent responses, for example by discarding query-key interactions and relying solely on intra-value correlations. Inspired by these approaches, we replace the original query-key attention $\text{Attn}_{qk}$ with a value-value attention $\text{Attn}_{vv}$ and remove the residual connections:

$$\text{Attn}_{vv}(X) = \text{Softmax}\left(\frac{VV^\top}{\sqrt{d_v}}\right)V. \qquad (3)$$

This adjustment simplifies the learning of local feature consistency by allowing the attention to operate solely on the value embeddings, which avoids global semantic bias induced by query-key interactions and promotes more spatially coherent region representations.

After the modified attention block, we discard the updated class token $Z'[0]$ and reshape the remaining patch embeddings $Z'[1 : h \times w]$ into a spatial feature map $X_{\text{patch}} \in \mathbb{R}^{C \times H \times W}$. These patch features can then be used to extract region-level representations, which will be further refined in the subsequent semantic distillation stage.

**Semantic Distillation Stage.** To further enhance the quality of CLIP patch features $X_{\text{patch}}$, we introduce the region-level feature extraction strategy and semantic-distillation mechanism specifically designed for open-vocabulary detection, as shown in Figure 3. Unlike existing distillation-based OVD methods that typically rely on uniform spatial sampling (Naeem et al., 2024; Wu et al., 2023b; 2024), we focus the learning process on semantically meaningful foreground areas by leveraging a class-agnostic RPN pre-trained on

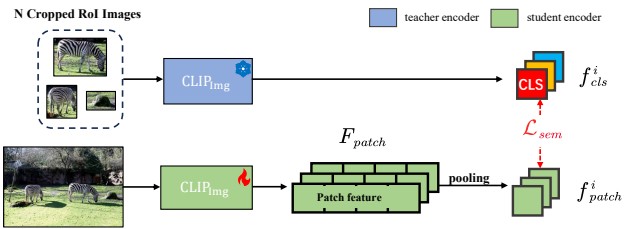

*Figure 3.* **Semantic Distillation.** Region proposals are encoded by a frozen CLIP teacher to obtain image-level [CLS] features. In parallel, student patch features are pooled within each region to form region embeddings, which are aligned with the teacher [CLS] representations via the semantic distillation loss $\mathcal{L}_{\text{sem}}$ to inject instance-level semantic cues.

base categories (Zhong et al., 2022). The RPN provides high-quality region proposals which can better align with object-centric representations required for the region-level feature extraction.

For each image $I$, we select the top 20 region proposals ranked by the RPN confidence score to reduce computational overhead:

$$R = \{r_i\}_{i=1}^N, \tag{4}$$

where $N$ denotes the number of selected region proposals (set to a maximum of 20 per image). The region $r_i$ is cropped from the original image to form a corresponding sub-image $I_i'$, which is independently passed to the teacher CLIP model to generate its image-level features $f_{\text{cls}}^i$ (i.e., the [CLS] token). At the same time, the student CLIP model processes the full image $I$ and outputs a patch feature map $F_{\text{patch}} \in \mathbb{R}^{C \times H \times W}$. Region features are then extracted from $F_{\text{patch}}$ using RoIAlign (Kaiming et al., 2017):

$$F_{\text{patch}} = \{f_{\text{patch}}^i\}_{i=1}^N, \quad f_{\text{patch}}^i \in \mathbb{R}^{1 \times C}, \tag{5}$$

where $f_{\text{patch}}^i$ represents the pooled feature vector corresponding to the $i$-th region proposal $r_i$, and $C$ is the channel dimension of the CLIP visual backbone. RoIAlign aggregates spatial features within each proposal to a region-level embedding with consistent dimensionality, enabling direct alignment with the teacher's image-level [CLS] representation during semantic distillation.

Finally, the region-level self-distillation loss aligns each student region feature with its teacher embedding using cosine similarity:

$$\mathcal{L}_{\text{sem}} = \frac{1}{n} \sum_{i=0}^n \left( 1 - \frac{f_{\text{patch}}^i \cdot f_{\text{cls}}^i}{||f_{\text{patch}}^i|| \cdot ||f_{\text{cls}}^i||} \right). \tag{6}$$

This distillation encourages the student's regional patch representations to mimic the stronger semantics encoded by the teacher's image-level embeddings, thereby enhancing the discriminability of region-level features crucial for open-vocabulary detection.

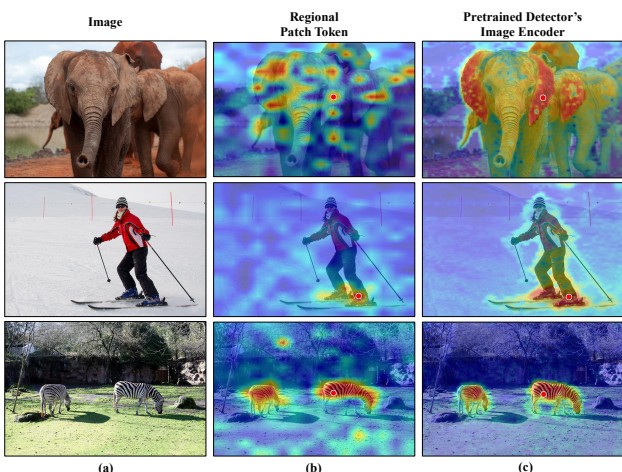

*Figure 4.* Visualization of token-level similarity maps from different feature representations.

**Spatial Distillation Stage.** Although the first distillation stage substantially enhances CLIP's semantic discriminability for better aligning with category cues, the resulting patch representations still inherit the background-biased behavior of the original CLIP backbone. As shown in Figure. 4(b), these features lack stable spatial sensitivity and coherent localization structure, making it difficult for downstream detectors to capture boundary-aware and shape-sensitive object signals. To overcome this limitation, we propose the second distillation stage to transfer spatial knowledge from a pre-trained detector that naturally encodes structural cues.

Following the standard open-vocabulary detection (OVD) protocol, we adopt a DINO-style transformer detector trained solely on the base categories $\mathcal{C}_{\text{base}}$ of the COCO dataset. A frozen CLIP image encoder is integrated as the visual backbone to provide open-vocabulary representations, while the detector is optimized with a standard detection loss:

$$\mathcal{L}_{\text{det}} = \mathcal{L}_{\text{cls}}(\hat{p}^c, p^c) + \lambda \cdot \mathcal{L}_{\text{box}}(\hat{b}^c, b^c), \tag{7}$$

where, $p^c$ and $b^c$ denote the ground-truth class label and bounding box for category $c$, respectively. The symbols $\hat{p}^c$ and $\hat{b}^c$ represent the predicted classification score and bounding box that are matched to the ground-truth instance via the Hungarian matching algorithm. This ensures the detector learns localization and shape priors without accessing novel-class annotations. After training, we extract spatially well-aligned features that serve as teacher signals to guide CLIP's patch features toward more localization-friendly representations.

As illustrated in Figure. 4(c), the similarity map in the image encoder of the pre-trained detector exhibits strong object-centric behavior, accurately highlighting object contours and local details. Our objective is to inject these spatial priors into CLIP's patch features so that they can simultaneously benefit from stronger semantics and more consistent

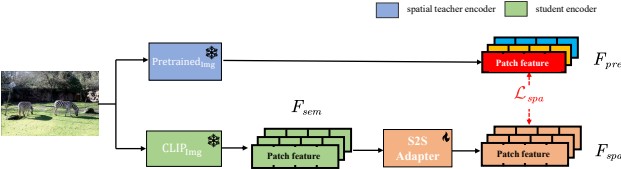

*Figure 5.* **Spatial Distillation.** A pre-trained detector serves as a spatial teacher to provide localization-aware supervision. In parallel, the S2S adapter transforms frozen CLIP features into spatial representations, which are aligned with the teacher's patch features via the spatial distillation loss $\mathcal{L}_{spa}$ to preserve fine-grained structural cues for accurate localization.

localization.

However, directly applying spatial distillation to the self-distilled CLIP features would compromise the semantic consistency achieved in the first stage, as semantic and spatial signals might interfere. To avoid this conflict, as shown in Figure 5, we design a Semantic-to-Spatial (S2S) Adapter, a lightweight convolutional transformation applied only to patch features. This module allows the student model to absorb pre-training encoder's structural cues while preserving the semantic alignment previously established.

During the spatial distillation stage, we freeze the CLIP backbone that has been optimized in the semantic distillation phase and only train the lightweight convolutional S2S Adapter. Given the semantically distilled regional patch representation $F_{sem} \in \mathbb{R}^{C \times H \times W}$, the S2S Adapter transforms semantically aligned features into spatially aligned features:

$$F_{spa} = \text{S2S}\left(F_{sem}\right), \qquad (8)$$

where S2S is implemented as a Conv-BN-ReLU block:

$$\text{S2S}\left(F_{sem}\right) = \sigma\left(\text{BN}\left(\text{Conv}_{3\times3}\left(F_{sem}\right)\right)\right), \qquad (9)$$

given the input semantic feature, it is first processed by a $3 \times 3$ convolution layer $\text{Conv}_{3\times3}(\cdot)$ to transform the feature from semantically aligned space into spatially aligned space, followed by batch normalization $\text{BN}(\cdot)$ and a sigmoid activation $\sigma(\cdot)$. Despite its minimal parameter cost, the adapter encourages spatial consistency in CLIP's patch features, which benefits downstream localization tasks.

To transfer pre-trained detector's spatial knowledge into the model, we propose a spatial-distillation mechanism. Given an input image $I$, we extract the image-level patch feature $F_{pre}$ from the teacher encoder of the pre-trained detector and the spatially adapted feature map $F_{spa}$ from the S2S adapter in the student CLIP. We compute patch-wise relational similarities between all token pairs for both the pre-trained teacher encoder and the student CLIP model, denoted as $R^{\text{PRE}}$ and $R^{\text{CLIP}}$, respectively. The student model is then encouraged to reproduce the structural correlations encoded by the teacher encoder

$$R^{\text{CLIP}} = \hat{F}_{spa}\hat{F}_{spa}^{\top}, \quad \hat{F}_{spa} = \frac{S2S(F_{sem})}{||S2S(F_{sem})||_2}, \qquad (10)$$

$$R^{\text{PRE}} = \hat{F}_{pre}\hat{F}_{pre}^{\top}, \quad \hat{F}_{pre} = \frac{F_{pre}}{||F_{pre}||_2}, \qquad (11)$$

where $\hat{X}$ denote L2-normalized patch embeddings, both relational matrices lie in the range $[-1, 1]^{M \times M}$; the number of patch tokens are $M = H \times W$. Consequently, we align the correlation patterns of student and teacher using the following loss:

$$\mathcal{L}_{spa} = \frac{1}{M^2}\left\|R^{\text{PRE}} - R^{\text{CLIP}}\right\|_F^2, \qquad (12)$$

where $\|\cdot\|_F$ denotes the Frobenius-norm, $R^{\text{PRE}}$ and $R^{\text{CLIP}}$ are the teacher and student correlation matrices, respectively, both of size $M \times M$. This Frobenius-norm formulation measures the global discrepancy between the two relational structures, encouraging the student model to faithfully reproduce pre-trained encoder's object-centric spatial organization across all patch relationships.

### 3.3. Location Quality Estimation Head

In most open-vocabulary detectors, each predicted proposal is assigned an objectness confidence score, which reflects the likelihood that the region corresponds to a foreground object, regardless of its semantic category. However, relying solely on objectness scores often fails to account for the precise alignment of predicted bounding boxes, particularly for unseen classes. To enhance spatial discrimination on novel categories, we introduce a Location Quality Estimation Head (LQEH) that explicitly learns to regress the localization quality of each detection proposal.

Specifically, we first employ the standard Hungarian matching algorithm to pair predicted proposals with corresponding ground-truth boxes. For each matched pair, the Intersection-over-Union (IoU), denoted as $s_i$, is calculated to serve as a continuous measure of spatial precision. We do not utilize $s_i$ directly as the supervision signal for training the LQEH to estimate localization quality. Instead, we modify the loss by replacing the fixed target label with a transformed IoU signal $f_1(s_i) = \varepsilon(s_i^2)$, which $\varepsilon$ is a transformation of the raw IoU values to dampen fluctuations and narrow the dynamic range, effectively preventing training instability and avoiding degenerate solutions. Consequently, the supervision loss for the LQEH is formulated as:

$$\mathcal{L}_{\text{IoU}} = \sum_{i=1}^{N_{\text{pos}}} |f_1(s_i) - p_i|^{\gamma} \cdot \text{BCE}\left(p_i, f_1(s_i)\right)$$
$$+ \sum_{i=1}^{N_{\text{neg}}} p_i^{\gamma} \cdot \text{BCE}\left(p_i, 0\right), \qquad (13)$$

where $p_i$ denotes the predicted confidence, and $\gamma$ is the standard Focusing Parameter to control the rate at which easy examples are down-weighted, allowing the model to

focus more on hard examples during training (Lin et al., 2017).

Building upon this localization-aware confidence, we perform class-agnostic bipartite matching (Wu et al., 2023c) between ground-truth boxes and model predictions. For an image with ground-truth set $Y = \{y^i\}_{i=0}^{N}$, and the prediction set $\hat{Y} = \{\hat{y}^i\}_{i=0}^{M}$, we identify the optimal one-to-one assignment by solving:

$$\hat{\sigma} = \text{HM}\left(\mathcal{L}_{\text{cost}}, Y, \hat{Y}\right), \tag{14}$$

where HM denotes the Hungarian matching algorithm. The cost function $\mathcal{L}_{\text{cost}}$ is defined as:

$$\mathcal{L}_{\text{cost}}\left(y^i, \hat{y}^j\right) = \mathcal{L}_{\text{match}}\left(y_p^i, \hat{y}_p^j\right) + \mathcal{L}_{\text{box}}\left(y_b^i, \hat{y}_b^j\right), \tag{15}$$

where $\mathcal{L}_{\text{match}}$ is implemented using the focal loss (Lin et al., 2017), while $\mathcal{L}_{\text{box}}$ combines an $L_1$ term with a GIoU loss (Zhai et al., 2020). Here, $y_p^i$ and $y_b^i$ denote the ground-truth class label and bounding box of the $i$-th object, respectively; while $\hat{y}_p^j$ and $\hat{y}_b^j$ represent the predicted classification score and bounding box of the $j$-th query, which are used to compute the matching cost prior to Hungarian assignment.

Importantly, this formulation performs matching at the image level without relying on any fixed category label, thus aligning naturally with the class-agnostic nature of open-vocabulary detection. Finally, the loss for OVD detector is defined as follows:

$$\mathcal{L} = \sum_{c \in C_{\text{base}}} \mathcal{L}_{\text{match}}(p^c, \hat{p}_{\hat{\sigma}_c}^c) + \mathcal{L}_{\text{box}}(b^c, \hat{b}_{\hat{\sigma}_c}^c) + \beta \mathcal{L}_{\text{IoU}}(s^c, \hat{s}_{\hat{\sigma}_c}^c), \tag{16}$$

where $C_{\text{base}}$ denotes the set of annotated base classes, $\beta$ denotes the IoU loss weight, $\hat{\sigma}_c$ denotes the optimal assignment obtained by the Hungarian algorithm. The Hungarian matching establishes a one-to-one correspondence between ground-truth objects and predictions, ensuring that each ground-truth instance is paired with a unique prediction for subsequent loss computation. Accordingly, $\hat{p}_{\hat{\sigma}_c}^c$, $\hat{b}_{\hat{\sigma}_c}^c$, and $\hat{s}_{\hat{\sigma}_c}^c$ denote the predicted classification score, bounding box, and IoU score corresponding to the matched prediction, respectively. The terms $p^c$, $b^c$, and $s^c$ represent the ground-truth class label, bounding box, and IoU target for category $c$.

# 4. Experiments

## 4.1. Datasets And Implementation Details

We conduct experiments on two widely used open- vocabulary detection benchmarks, OV-COCO(Lin et al., 2014) and OV-LVIS(Gupta et al., 2019). Due to space limitations, detailed descriptions of the datasets, evaluation metrics are provided in the Appendix.

**Details About Distillation Learning.** We use the base categories of OV-COCO to train the RPN, enabling it to generate candidate foreground proposals. During training, all layers of the student CLIP image encoder are trainable. We adopt the AdamW optimizer with a learning rate of $3 \times 10^{-6}$ and a weight decay of $1 \times 10^{-4}$, using a batch size of 1 per GPU. In the spatial distillation stage, a DINO-DETR (Oquab et al., 2023) encoder pre-trained on the OV-COCO base classes is employed as the teacher model to distill spatial knowledge into the S2S adapter. While distilling, only the S2S adapter is optimized. This stage uses the AdamW optimizer with a learning rate of $5 \times 10^{-5}$ and a batch size of 32 per GPU. All training procedures are performed on 4 NVIDIA RTX 3090 GPUs.

**Details About Detection.** Our OSSD framework is built upon the closed-set DINO-DETR architecture. In line with prior open-vocabulary detection approaches (Wang et al., 2025; Wu et al., 2023c), to extend it to the open-vocabulary setting, we reformulate the decoder to perform conditioned matching between object queries and text embeddings, enabling category prediction without a fixed closed-set classifier. The detector employs 1,000 learnable object queries and a Transformer backbone consisting of 6 encoder layers and 6 decoder layers. To ensure a fair comparison with pseudo-label-based methods, we adopt a self-training strategy (Wang et al., 2025; Wu et al., 2023c) to generate pseudo labels for model training.

The model is trained with a per-GPU batch size of 4 using the AdamW optimizer, with a learning rate of $1 \times 10^{-4}$ and a weight decay of $1 \times 10^{-4}$. During training, the Hungarian matching algorithm assigns cost weights of 2.0, 5.0, and 2.0 to the objectness score, bounding box regression, and GIoU terms, respectively. For the Location Quality Estimation Head (LQEH), the localization quality supervision is weighted by a factor of $\beta = 3.0$.

## 4.2. Open-Vocabulary Detection Results

Table 1 and Table 2 summarize the performance of OSSD on the OV-COCO and OV-LVIS benchmark. To ensure a fair comparison, we explicitly report whether each method leverages extra training data, requires annotations of novel classes, and the backbone architecture employed, as these factors vary across methods and have a substantial impact on performance.

As shown in Table 1, OSSD consistently outperforms existing methods across different backbone settings, without relying on additional datasets or novel-class annotations. With a ViT-B/16 backbone, OSSD achieves 43.7 $\text{AP}_{50}^{\text{Novel}}$ on novel classes, surpassing the previous state-of-the-art ViT-base method CLIPSelf by 6.1 $\text{AP}_{50}^{\text{Novel}}$. When adopting the stronger ViT-L/14 backbone, OSSD further improves performance to 49.2 $\text{AP}_{50}^{\text{Novel}}$, maintaining a lead

*Table 1.* Results on the OV-COCO benchmark. 'B', 'L', and 'H' in ViT-based methods denote base, large, and huge model sizes, respectively. '/16' and '/14' indicate the input image downsampling ratios. † denotes methods that use novel class names during training. Different variants of our method are also reported, where '+Encoder' and '+DINOv2' denote the teacher models used in the spatial distillation stage.

| Model | Extra Data | Backbone | $AP_{50}^{Novel}$ |
|---|---|---|---|
| BARON-KD (Wu et al., 2023a) | – | RN50 | 34.0 |
| CORA (Wu et al., 2023c) | – | RN50 | 35.1 |
| CLIM (Wu et al., 2024) | COCO Captions | RN50 | 36.9 |
| SAS-Det (Zhao et al., 2024) | – | RN50 | 37.4 |
| OV-DQUO (Wang et al., 2025) | – | RN50 | 39.2 |
| RegionCLIP (Zhong et al., 2022) | CC3M | RN50×4 | 39.3 |
| CORA (Wu et al., 2023c) | – | RN50×4 | 41.7 |
| CORA+† (Wu et al., 2023c) | COCO Captions | RN50×4 | 43.1 |
| OV-DQUO (Wang et al., 2025) | – | RN50×4 | 45.6 |
| ViLD† (Gu et al., 2021) | – | ViT-B/32 | 27.6 |
| OV-DETR† (Zang et al., 2022) | – | ViT-B/32 | 29.4 |
| CLIPSelf (Wu et al., 2023b) | – | ViT-B/16 | 37.6 |
| PromptDet† (Song & Bang, 2023) | COCO | ViT-L/16 | 30.6 |
| CFM-ViT (Kim et al., 2023a) | – | ViT-L/16 | 34.1 |
| BIND (Zhang et al., 2024) | – | ViT-L/16 | 41.5 |
| CLIPSelf (Wu et al., 2023b) | – | ViT-L/14 | 44.3 |
| OSSD+DETR_Encoder (Ours) | – | ViT-B/16 | 43.7 |
| OSSD+DINOv2 (Ours) | – | ViT-B/16 | 44.9 |
| OSSD+DETR_Encoder (Ours) | – | ViT-L/14 | 49.2 |
| OSSD+DINOv2 (Ours) | – | ViT-L/14 | **52.4** |

over CLIPSelf of 4.9 $AP_{50}^{Novel}$. For ViT-based backbones, OSSD demonstrates clear advantages. Under the RN50×4 setting, OSSD achieves a 3.6 $AP_{50}^{Novel}$ improvement over OV-DQUO, the strongest prior method, highlighting the robustness of the proposed framework across both convolutional and transformer-based architectures.

Beyond the standard spatial distillation setup, where a DETR encoder pre-trained on OV-COCO base classes serves as the teacher, we further explore the generality of the proposed spatial distillation strategy by incorporating alternative foundation models as teachers. In particular, additional distillation experiments using DINOv2 (Oquab et al., 2023) provide stronger spatial priors, further boosting the performance to 52.4 $AP_{50}^{Novel}$. The consistent gains observed across these settings validate the effectiveness and robustness of the proposed spatial distillation mechanism, indicating that OSSD does not depend on a specific teacher architecture or training paradigm to acquire transferable spatial knowledge.

We further extend our evaluation to the OV-LVIS benchmark, as shown in Table 2. Since LVIS contains a substantially larger and more diverse set of categories than COCO (1203 vs. 80), training spatial teachers on LVIS base classes becomes considerably more challenging. Therefore, on OV-LVIS, we focus on the backbone setting using ViT-L/14, which allows us to assess the scalability and practicality of OSSD under more realistic large-vocabulary scenarios. Under this setting, OSSD achieves 40.5 $mAP_r$ on novel categories. Moreover, when extending the spatial teacher to DINOv2, the performance is further improved to 41.8 $mAP_r$.

*Table 2.* Results on the OV-LVIS benchmark. 'B', 'L', and 'H' in ViT-based methods denote base, large, and huge model sizes, respectively. '/16' and '/14' indicate the input image downsampling ratios. † denotes methods that use novel class names during training. Different variants of our method are also reported, where '+Encoder' and '+DINOv2' denote the teacher models used in the spatial distillation stage.

| Method | Extra Dataset | Backbone | $mAP_r$ |
|---|---|---|---|
| RegionCLIP† (Zhong et al., 2022) | CC3M | RN50×4 | 22.0 |
| OV-SAM (Zhong et al., 2022) | – | RN50×16 | 24.0 |
| CORA+† (Wu et al., 2023c) | IN-21K | RN50×4 | 28.1 |
| SAS-Det (Zhao et al., 2024) | – | RN50×4 | 29.1 |
| CLIM (Wu et al., 2024) | COCO Captions | RN50×64 | 32.3 |
| F-VLM (Kuo et al., 2022) | – | RN50×64 | 32.8 |
| RTGen (Chen et al., 2024) | Conceptual Captions | Swin-B | 30.2 |
| Detic† (Zhou et al., 2022) | Conceptual Captions | Swin-B | 33.8 |
| ProxyDet (Jeong et al., 2024) | COCO Captions | Swin-B | 36.7 |
| BIND (Zhang et al., 2024) | – | ViT-L/16 | 32.5 |
| CFM-ViT (Kim et al., 2023a) | – | ViT-L/14 | 33.9 |
| RO-ViT (Kim et al., 2023b) | – | ViT-H/16 | 30.6 |
| CLIPSelf (Wu et al., 2023b) | – | ViT-L/14 | 34.9 |
| CoDet (Ma et al., 2023) | COCO Captions | ViT-L/14 | 37.0 |
| OV-DQUO (Wang et al., 2025) | – | ViT-L/14 | 39.3 |
| OSSD+DETR_Encoder (Ours) | – | ViT-L/14 | 40.5 |
| OSSD+DINOv2 (Ours) | – | ViT-L/14 | **41.8** |

### 4.3. Ablation Study

To analyze the contribution of each component, we conduct an ablation study on the OV-COCO benchmark, with results summarized in Table 3. Our baseline is an unmodified DINO-DETR detector with a vanilla ViT-L/14 backbone, which achieves 38.3 $AP_{50}^{Novel}$ on novel categories when trained with self-generated pseudo labels (Wang et al., 2025; Wu et al., 2023c). Then, we individually activate Spatial Distillation, Semantic Distillation, and LQEH to evaluate their isolated effects. As shown in configurations A–C, enabling Spatial Distillation alone improves novel AP to 42.3 $AP_{50}^{Novel}$, while Semantic Distillation yields a larger gain of 6.7 $AP_{50}^{Novel}$, reaching 45.0. Incorporating LQEH in isolation also brings a notable improvement, achieving 41.7 $AP_{50}^{Novel}$.

We further examine the complementary effects of these components. Combining Spatial and Semantic Distillation (configuration D) leads to a substantial performance boost, elevating novel AP to 47.8 $AP_{50}^{Novel}$, which highlights their strong synergy. While Semantic Distillation significantly enhances region-level semantic discriminability, it alone does not fully address the background-biased and spatially

*Table 3.* Ablation study on the main effective components of OSSD.

| | Spatial Distillation | Semantic Distillation | LQEH | $AP_{50}^{Novel}$ |
|---|---|---|---|---|
| baseline | - | - | - | 38.3 |
| A | ✓ | ✗ | ✗ | 42.3 |
| B | ✗ | ✓ | ✗ | 45.0 |
| C | ✗ | ✗ | ✓ | 41.7 |
| D | ✓ | ✓ | ✗ | 47.8 |
| E | ✓ | ✓ | ✓ | 49.2 |

unstable representations inherited from the CLIP backbone. Spatial Distillation effectively compensates for this limitation by enforcing spatial sensitivity and coherent localization structure, thereby enabling the detector to capture boundary-aware and shape-sensitive object cues.

Finally, integrating all three components (configuration E) delivers the best performance, achieving 49.2 $AP_{50}^{Novel}$ on novel categories. These results demonstrate that all modules contribute positively to novel-class detection, with Semantic Distillation serving as the primary driver and Spatial Distillation acting as a crucial complement, while LQEH further provides consistent gains when coupled with distillation-based learning.

**Region-Level Classification.** Since classification is explicitly decoupled from localization in our framework, the CLIP-based region classifier can be evaluated independently through a region-level classification task. After semantic distillation on the COCO dataset, we assess the distilled CLIP on the OV-COCO test set over both base and novel categories, reporting Top-1 mean accuracy on ground-truth bounding boxes and systematically analyzing the impact of different attention designs and prompting strategies.

As shown in Table 4, vanilla CLIP exhibits limited discriminability at the region level. Although region prompting improves performance on base categories, its base-only optimization generalizes poorly to novel classes across both RN50x4 and ViT-L/14 backbones, revealing a pronounced bias toward base categories.

In response to the attention structure modifications introduced in Section 3.2, we observe that, for ViT-L/14, altering the last attention layer formulation consistently leads to performance improvements. Among the variants, value-value (v-v) attention performs best overall. Moreover, applying self-distillation across all layers yields more balanced gains, particularly on novel categories, suggesting that preserving global semantic consistency is crucial for robust open-

vocabulary generalization.

## 5. Conclusions

In this paper, we propose OSSD, an object-level semantic and spatial distillation framework for open-vocabulary object detection. We observe that existing distillation-based OVD methods often improve semantic alignment at the expense of spatial precision, limiting localization performance on novel objects. To address this issue, OSSD adopts a two-stage distillation strategy that jointly preserves semantic discriminability and spatial coherence. In addition, we introduce a Location Quality Estimation Head (LQEH) to explicitly model class-agnostic localization quality, improving the identification of well-localized novel instances. Experiments on OV-COCO and OV-LVIS demonstrate that OSSD consistently outperforms state-of-the-art methods, with notable gains in both recall and precision on novel categories. We believe this work provides a principled perspective on balancing semantic alignment and spatial fidelity in vision-language detectors and opens new directions for designing more robust open-vocabulary detection frameworks.

## Acknowledgements

We would like to thank anonymous reviewers for their suggestions and comments sincerely. This work is supported by the National Natural Science Foundation of China (62573393, 62373338).

## Impact Statement

This paper presents work whose goal is to advance the field of Machine Learning. There are many potential societal consequences of our work, none which we feel must be specifically highlighted here.

*Table 4.* Sensitivity Analysis for Region-Level Classification.

| Backbone | Method | Layer | Attention | Score (Top1 Acc) | | |
|---|---|---|---|---|---|---|
| | | | | **Base** | **Novel** | **All** |
| **RN50x4** | Vanilla CLIP | - | - | 49.8 | 64.6 | 51.9 |
| | + Region Prompt | Pooling | - | 80.7 | 65.2 | 78.5 |
| **ViT-L/14** | Vanilla CLIP | - | - | 38.8 | 54.2 | 40.9 |
| | + Change the last Attention | final | k-k | 66.1 | 66.0 | 66.1 |
| | + Change the last Attention | final | q-q | 68.2 | 69.0 | 68.3 |
| | + Change the last Attention | final | v-v | 69.0 | 71.9 | 69.4 |
| | + Token Prompt | final | v-v | 81.1 | 72.2 | 79.9 |
| | + Token Prompt | Last 5 | v-v | 88.2 | 73.6 | 84.8 |
| | + Semantic Distillation | Last 3 | v-v | 70.4 | 74.5 | 71.0 |
| | + Semantic Distillation | all | v-v | **76.3** | **80.1** | **77.2** |

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

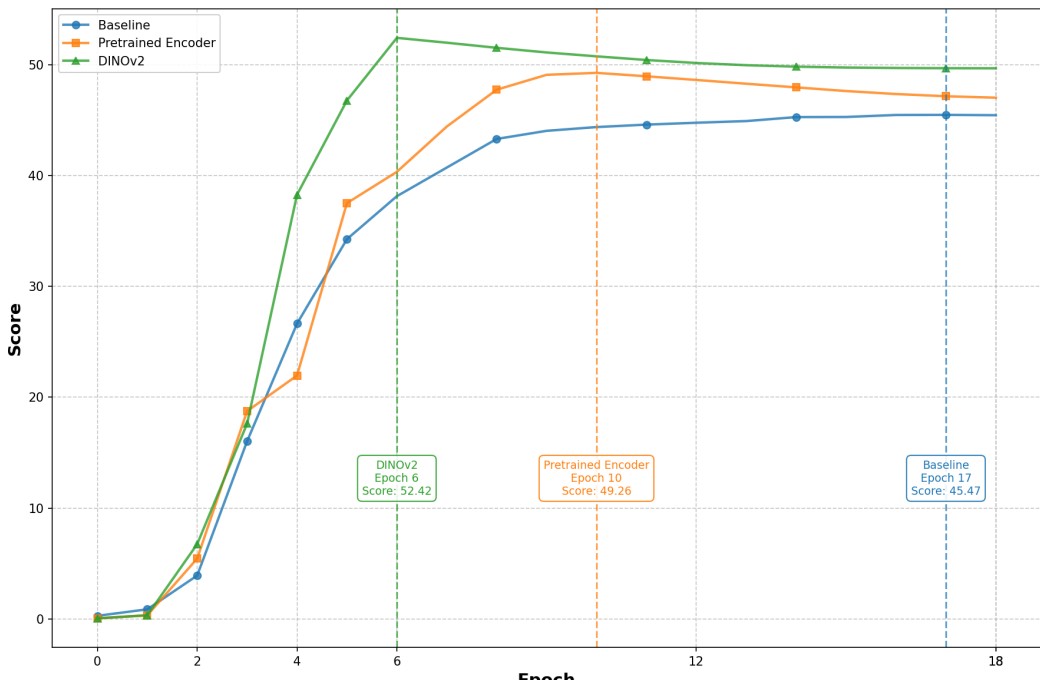

*Figure 6.* Convergence behavior of novel $AP_{50}$ across training epochs. Spatial distillation significantly accelerates convergence and improves peak performance on novel categories, especially when using a stronger teacher model such as DINOv2.

## A. Datasets And Evaluation Metrics Details

**OV-COCO Benchmark.** The original 80 categories in the COCO dataset are partitioned into a set of base categories and a disjoint set of novel categories, following the standard evaluation protocol. The detector is trained exclusively on the base split, which includes 48 categories with 107,761 images and 665,387 annotated objects. Performance is assessed on the validation set that covers both base and novel categories, consisting of 4,836 images and 33,152 object instances. We report the box AP at IoU threshold 0.5 of novel categories, which is denoted as $AP_{50}^{Novel}$.

**LVIS Benchmark.** During training, only frequent and common categories are used, totaling 866 classes (461 common and 405 frequent), with 100,170 images and 1,264,883 instances. The trained model is then evaluated on the LVIS validation set, which contains annotations from frequent, common, and rare categories, spanning 19,809 images and 244,707 instances. For OV-LVIS, we report mean Average Precision ($mAP_r$) of rare categories on IoUs from 0.5 to 0.95 ($mAP_r$).

## B. Convergence Behavior

**Convergence Behavior of Beyond Training Epochs.** Beyond final performance, we further investigate the convergence behavior of different methods by analyzing the evolution of novel $AP_{50}^{Novel}$ across training epochs, as shown in Figure 6. The state-of-the-art (SOTA) baseline exhibits relatively slow convergence, reaching 44 $AP_{50}^{Novel}$ at epoch 10 and gradually improving to its peak performance of 45.5 $AP_{50}^{Novel}$ after 17 epochs.

In contrast, our method converges significantly faster when Spatial Distillation is applied at the encoder level. Under this setting, the model attains its optimal performance of 49.2 $AP_{50}^{Novel}$ within only 10 epochs. The advantage becomes even more pronounced when a stronger teacher model (DINOv2) is employed for spatial distillation, enabling the detector to reach a substantially higher peak of 52.4 $AP_{50}^{Novel}$ in merely 6 epochs.

These observations indicate that Spatial Distillation injects strong spatial priors and structured feature representations at early training stages, which stabilizes optimization dynamics, accelerates convergence, and ultimately leads to superior novel-category detection performance.

*Table 5.* Sensitivity analysis of IoU-based supervision functions in LQEH.

|  | $f_1(s)$ | $AR_{50}^{Novel}$ | $AP_{50}^{Novel}$ | $AP_{50}^{All}$ |
|---|---|---|---|---|
| baseline | - | 71.1 | 45.6 | 48.1 |
| 1 | $s^{0.5}$ | 71.7 | 47.9 | 50.7 |
| 2 | $s$ | 74.8 | 48.7 | 52.1 |
| **3** | $s^2$ | **76.6** | **49.2** | **52.3** |
| 4 | $s^3$ | 75.8 | 49.0 | 51.5 |
| 5 | $(e^s - 1)/(e - 1)$ | 76.2 | 49.0 | 51.9 |
| 6 | $\sin(s \times \pi/2)$ | 74.0 | 48.3 | 49.1 |

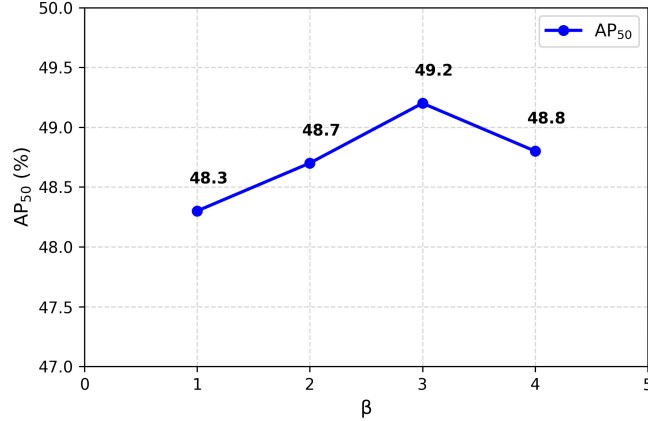

*Figure 7.* Sensitivity analysis of loss weight $\beta$ in LQEH.

# C. Hyperparameter Selection.

**IoU-based Supervision Function Selection.** We further investigate the sensitivity of our model to the form of IoU-based supervision $f_1(s_i)$ and the weighting factor $\beta$ in the Location Quality Estimation Head (LQEH).

As illustrated in Table 5, different transformations of the IoU signal exhibit distinct behaviors. Linear IoU supervision provides limited discrimination between medium- and high-quality regions, whereas the exponential mapping excessively amplifies high-IoU samples and leads to unstable optimization. In contrast, the squared IoU function offers a smoother yet more discriminative weighting scheme, effectively emphasizing well-localized proposals while suppressing noisy regions. This observation motivates our choice of $f_1(s_i) = \varepsilon(s_i^2)$, which consistently delivers the best AR and AP on novel categories and facilitates robust separation of novel instances from background noise.

**Loss Weight Selection.** Figure 7 studies the influence of the LQEH loss weight $\beta$. While the detector exhibits stable performance across different choices of $\beta$, a moderate weighting ($\beta = 3.0$) leads to the strongest gains on novel categories. This suggests that appropriately balancing objectness and localization-aware supervision helps the model better distinguish well-localized novel objects from background noise.

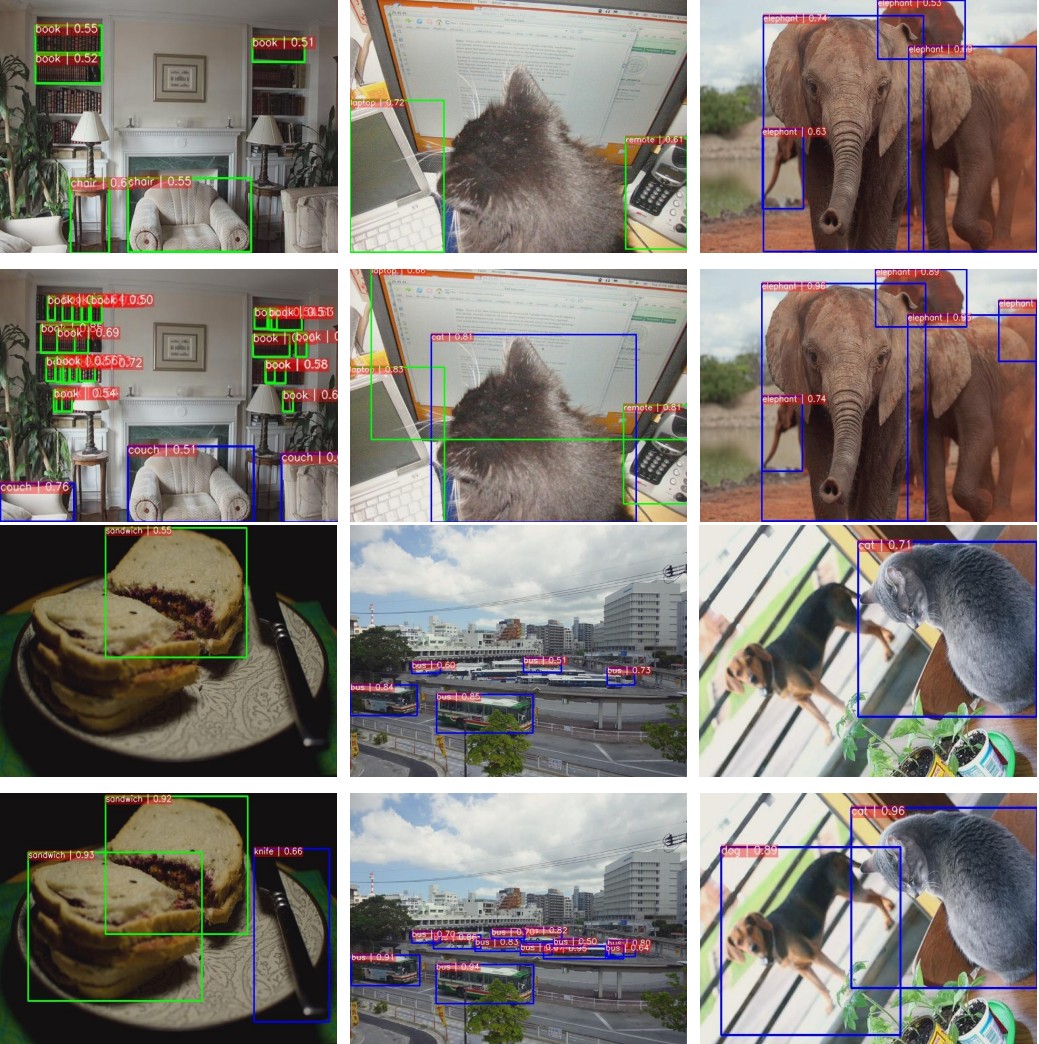

*Figure 8.* Visualization of detection results for the OV-COCO benchmark. Blue boxes represent novel categories and green boxes indicate base categories.

## D. Visualization Analysis

We compare the detection results of the baseline model (top row) and our OSSD (bottom row) on the OV-COCO validation set, as shown in Figure 8. While the baseline often produces incomplete detections or misses novel objects, OSSD successfully identifies multiple novel categories with more accurate and complete bounding boxes. In particular, OSSD demonstrates improved localization and confidence on challenging novel instances such as couch, knife, cat, and dog, highlighting its effectiveness in reducing base-class bias and enhancing open-vocabulary generalization.

As shown in Figure 9, given a reference patch token (highlighted in red), we compute its cosine similarity with all other patch tokens and visualize the resulting patch-wise similarity map, where warmer colors indicate higher similarity. After applying the spatially distilled S2S adapter, the similarity maps not only preserve the strong semantic correlations induced by semantic distillation—highlighting different instances of the same category—but also exhibit enhanced spatial coherence. This demonstrates that our model effectively integrates semantic consistency with spatial awareness, enabling it to capture spatially coherent and semantically meaningful correspondences.

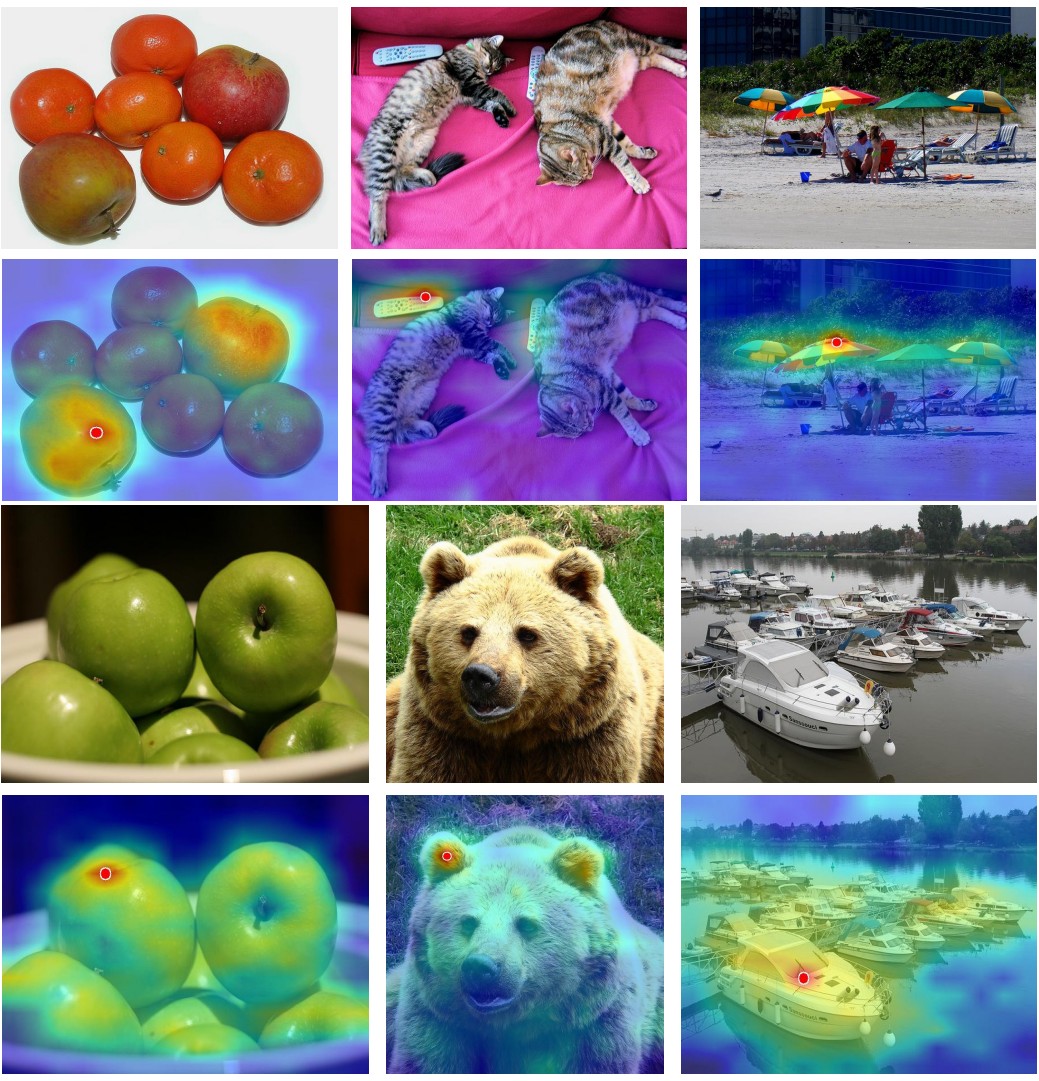

*Figure 9.* Visualization of token-level similarity maps from different feature representations. All maps are obtained by computing the cosine similarity between a reference token (highlighted in red) and all other visual tokens, and visualizing the resulting patch-wise similarity maps.

