# OpenReview forum: "Object-level Semantic and Spatial Distillation for Open Vocabulary Detection"
_ICML.cc/2026/Conference — ICML 2026 regular_

### Official Review · Reviewer_DRKT · 2026-03-08

**Soundness:** 3
**Presentation:** 3
**Significance:** 3
**Originality:** 3
**Overall Recommendation:** 4
**Confidence:** 4

**Summary:**

This paper presents an elegant idea for open-vocabulary object detection by distilling rich semantic knowledge from pre-trained CLIP and spatial priors from a pre-trained detector. It achieves highly impressive results on both OV-COCO and OV-LVIS.

**Compliance With Llm Reviewing Policy:**

Affirmed.

**Final Justification:**

This paper meets the quality standard for ICML, and I recommend accepting it for publication.

**Key Questions For Authors:**

1. Despite the experimental results shown in Table 4, the Value-Value Attention defined in Eq. 3 lacks sufficient theoretical validation.
The authors should further clarify how the method avoids global semantic bias.
2. There are several recent state-of-the-art methods that also rely on pre-trained models and achieve stronger performance on OV-COCO, such as VLDet (Enhancing Open-Vocabulary Object Detection through Multi-Level Fine-Grained Visual-Language Alignment) and C3-OWD ( C3-OWD: A Curriculum Cross-modal Contrastive Learning Framework for Open-World Detection).  Though a like-for-like numerical comparison may not be feasible, the authors are strongly encouraged to provide a discussion and analysis of these related works.

**Limitations:**

Yes

**Strengths And Weaknesses:**

Strengths
1. The motivation of the proposed method is theoretically sound from the perspective of object detection.
2. The experimental results are highly impressive and the proposed method is supposed to benefit the research community of open-vocabulary object detection.
3. The paper is well organized and clearly presented, making it easy to follow and understand.

Weaknesses
1. The performance of the proposed method largely relies on the strong prior knowledge embedded in the pre-trained models. The authors are strongly encouraged to further discuss the originality and novelty of the proposed distillation pipeline, especially in comparison with existing methods that also leverage pre-trained models.

---

> ### Author Rebuttal · Authors · 2026-03-30
>
> We thank the reviewer for the valuable comments and insightful suggestions. We address each concern below.
>
> **W1**
>
> **Response:** We thank the reviewer for the insightful comment. While pre-trained models provide strong priors, our gains mainly come from the distillation pipeline that adapts CLIP to detection, rather than stronger pretraining.
>
> First, our primary comparisons follow the standard OVD setting, where the spatial teacher is a DETR-style encoder trained only on COCO base classes, ensuring no novel-class knowledge leakage. Under this fair setup, our method still achieves consistent improvements over strong baselines.
>
> Second, spatial distillation is designed to enhance spatial awareness (e.g., boundaries and localization), rather than transfer semantics, complementing CLIP’s weakness in dense prediction.
>
> We also report results with stronger teachers (e.g., DINOv2) only as supplementary analysis to demonstrate scalability, not for primary comparison.
>
> Finally, we include comparisons with recent methods relying on MLLMs or stronger pretraining. As shown in Table R1, our method still achieves superior performance on OV-COCO/OV-LVIS, which will be further discussed in the final version.
>
> **Table R1: Results on OV-COCO benchmark.**
>
> | Method | Pre-train Model | Require Novel Class | $AP_{50}^{Novel}$ |
> | :--- | :---: | :---: | :---: |
> | CoT-PL[1] | RN50 | ✓ | 43.4 |
> | CoT-PL[1]  | RN50x4 | ✓ | 47.8 |
> | VMCNet[2] | VMCNet-L | ✗ | 48.5 |
> | **OSSD+DETR Encoder** | ViT-B/16 | ✗ | 43.7 |
> | **OSSD+DINOv2** | ViT-B/16 | ✗ | 44.9 |
> | **OSSD+DETR Encoder** | ViT-L/14 | ✗ | 49.2 |
> | **OSSD+DINOv2** | ViT-L/14 | ✗ | 52.4|
>
> **Table R2: Results on OV-LVIS benchmark.**
>
> | Method | Pre-train Model | Require Novel Class | $mAP_{r}$ |
> | :--- | :---: | :---: | :---: |
> | CoT-PL[1] | RN50x4 | ✓ | 26.4 |
> | VMCNet[2] | VMCNet-L | ✗ | 38.4 |
> | **OSSD+DETR Encoder** | ViT-L/14 | ✗ | 40.5 |
> | **OSSD+DINOv2** | ViT-L/14 | ✗ | 41.8 |
>
> [1] Choi, H., Lim, Y., Shin, J., et al. Cot-pl: Visual chain-of-thought reasoning meets pseudo-labeling for open-vocabulary object detection. arXiv preprint arXiv:2510.14792, 2025.
>
> [2] Gao, X., Dai, Y., Qiu, B., et al. Modulating cnn features with pre-trained vit representations for open-vocabulary object detection. arXiv preprint arXiv:2501.16981, 2025.
>
> **Q1**
>
> **Response:** We thank the reviewer for this insightful question. We clarify that the Value-Value (VV) attention in Eq. (3) is not our main novelty, but adopted from prior work (Page 4, Line 217), where it has been well studied.
>
> Our contribution is to clarify its role in mitigating CLIP’s global semantic bias. In standard Q-K attention, query and key are projected differently, which may cause inconsistent semantic associations. Combined with the lack of local supervision in CLIP, this often leads to attention focusing on irrelevant regions (the “reverse visualization” issue).
>
> In contrast, VV attention constructs the attention map using homogeneous features (V Vᵀ), enforcing stronger semantic consistency and more locally coherent interactions. This reduces spurious global correlations and improves spatial focus.
>
> Therefore, VV attention alleviates global semantic bias and benefits dense prediction tasks like OVD. While previously studied in segmentation, we validate its effectiveness in OVD, showing more stable and accurate localization (Table 4). We will further clarify this in the revision.
>
> **Q2**
>
> **Response:** We thank the reviewer for highlighting these recent works. As noted, our comparisons follow the standard OVD setting, where the spatial teacher is a DETR-style encoder trained only on COCO base classes, ensuring no additional novel-class knowledge and fair comparison.
>
> We have carefully reviewed VLDet and C3-OWD. These methods achieve strong performance on OV-COCO by leveraging additional large-scale pretraining or more advanced cross-modal alignment strategies. For instance, VLDet is pretrained on Objects365, which provides broader semantic coverage, making direct comparison less aligned with the standard OVD protocol.
>
> Notably, although VLDet performs well on sOV-COCO (58.7 mAP), its OV-LVIS performance (24.8 mAPr) is significantly lower than ours (40.5 mAPr). This suggests that methods relying on large-scale category-rich pretraining may not generalize as effectively to more diverse open-vocabulary settings.
>
> In contrast, our method does not rely on external semantic supervision. Instead, spatial distillation enhances localization (e.g., boundaries and spatial sensitivity) without transferring novel-class semantics, enabling stronger generalization on large-vocabulary datasets like LVIS.We will include a more detailed discussion of these differences in the revised manuscript.

---

> > ### Author Rebuttal · Reviewer_DRKT · 2026-04-03
> >
> > I am fine with the responses and inclined to accept the paper.

---

### Official Review · Reviewer_oYQS · 2026-03-11

**Soundness:** 3
**Presentation:** 3
**Significance:** 3
**Originality:** 3
**Overall Recommendation:** 5
**Confidence:** 4

**Summary:**

The submission proposes OSSD, a two-stage framework for open-vocabulary detection, which decouples semantic and spatial learning to avoid hurting localization when strengthening semantics. Stage 1 performs object-level semantic distillation: proposals from an RPN are cropped and encoded by a frozen CLIP to obtain [CLS] embeddings, while the student pools patch features via RoIAlign and aligns them with the teacher using a cosine loss. The last CLIP attention layer is modified to value–value attention (without residuals) to reduce global bias and improve local coherence. Stage 2 conducts spatial distillation from a detector encoder trained only on COCO base classes (or from DINOv2): the CLIP backbone is frozen and a lightweight S2S adapter is trained to match teacher–student patch correlation matrices, injecting boundary- and shape-aware priors. A class-agnostic Location Quality Estimation Head (LQEH) predict localization quality using a transformed IoU target with focal weighting, improving proposal ranking for novel objects.

**Compliance With Llm Reviewing Policy:**

Affirmed.

**Final Justification:**

Thanks the authors for the responses in detail. My concerns are addressed with evidences, like the computational overhead and dependence on base-supervised components. I tend to accept this submission. Thus, I raise my score to 5.

**Key Questions For Authors:**

1. How does the two-stage distillation scale in compute and memory, and what are the training or inference latency trade-offs, compared to single-stage or no-distillation baselines?

2. How robust is the proposd approach under distribution shifts (tiny/occluded/dense objects, non-COCO bases), and how much do RPN and spatial teacher biases affect novel-category performance?

3. What is the impact of the attention modification scope (last-layer value–value, residual removal) on text–image alignment and detection accuracy, and would alternative configurations perform better?

**Limitations:**

None. I have presented the limitation in the 'weakness' part.

**Strengths And Weaknesses:**

Strengths

1. Clear problem decomposition: decouples semantic and spatial learning to resolve the recognition–localization trade-off in OVD.

2. Solid technical design: object-level semantic distillation with CLIP [CLS] targets. Spatial distillation via patch–patch correlation matching. Lightweight S2S adapter avoids interference between stages.

3. Thoughtful CLIP modification: replacing the last attention with value–value attention to curb global bias and enhance local coherence.

4. Class-agnostic localization head (LQEH): improves ranking of novel objects, by predicting localization quality with a stabilized IoU target and focal weighting.

5. Strong and consistent empirical results: SOTA on OV-COCO and OV-LVIS across backbones; gains persist with different spatial teachers (DETR encoder, DINOv2).

Weaknesses

1. Computational overhead not quantified: two-stage training, teacher encoders, and correlation-matrix losses likely increase cost. Missing FLOPs/memory/wall-clock reporting and inference latency analysis.

2. Limited exploration of attention modification scope: only the last layer without residuals is studied. Broader ablations (layer depth, partial residuals) and effects on text–image alignment could strengthen claims.

3. Dependence on base-supervised components: RPN proposals and base-trained spatial teachers may embed dataset biases. Robustness under heavy domain shifts (tiny objects, dense scenes, long-tail) is underexplored.

4. LQEH comparison breadth:Lacks direct head-to-head with alternative quality-aware heads (e.g., GFL/Varifocal/Rank-DETR) or combinations with them.

5. External dat or /teacher variability: while different teachers are tried, the method’s behavior with weaker or noisy teachers and on non-COCO bases is not systematically analyzed.

6. Failure case analysis is light: more diagnostics on missed/false positives for novel categories would help guide deployment and further improvements.

7. Lacking discussion with 3D open-vocabulary detection [1,2] in the related work: what's the differences between the designs and will the proposed idea have potential to be applied in 3D domain?


[1] Yang Cao, Yihan Zeng, Hang Xu, and Dan Xu. Coda: Collaborative novel box discovery and cross-modal alignment for open-vocabulary 3D object detection. In NeurIPS, 2023.

[2] Pengkun Jiao, Na Zhao, Jingjing Chen, and Yu-Gang Jiang. Unlocking Textual and Visual Wisdom: Open-Vocabulary 3D Object Detection Enhanced by Comprehensive Guidance from Text and Image. In ECCV, 2024.

---

> ### Author Rebuttal · Authors · 2026-03-30
>
> We thank the reviewer for the valuable comments and insightful suggestions. We address each concern below.
>
> The detailed tables are provided at the following anonymous link: https://anonymous.4open.science/r/Rebuttal-sheets-30A0
>
> **W1 and Q1**
>
> **Response:** We thank the reviewer for the insightful suggestion. We provide a detailed analysis of the computational overhead of the multi-teacher framework and two-stage distillation, including FLOPs, training time, and memory (Tables R1–R2).Overall, although multi-teacher distillation adds some training overhead, it is well controlled without compromising efficiency, and inference cost remains nearly unchanged, ensuring practical deployment.
>
> **W2 and Q3**
>
> **Response:** We clarify that the Value-Value (VV) attention in Eq. (3) is adopted from prior works (Page 4, Line 217) and is not our primary novelty. Our contribution is to analyze its role in mitigating CLIP’s global semantic bias and extend it to OVD.
>
> VV attention uses homogeneous features (V-V^T), enforcing stronger semantic consistency and reducing the “reverse visualization” issue, where attention focuses on irrelevant regions. We apply VV attention only to the last layer, as higher layers contain more semantically aligned features with text, making them more suitable for improving localization. Due to space constraints, we focus on the most stable setting and will add further discussion in the revision.
>
> **W3**
>
> **Response:** We thank the reviewer for the insightful suggestion. Spatial distillation does not transfer novel-category semantics, but enhances spatial awareness (e.g., boundaries and localization). It is fully unsupervised, using only 5K randomly sampled COCO images and converging within 3 epochs, indicating it learns general spatial priors rather than dataset-specific bias.
>
> The RPN is class-agnostic and serves only as a proposal generator. Consistent gains with the same RPN on LVIS further suggest improvements are not due to proposal bias.
>
> Finally, we adopt a conservative distillation strategy (e.g., small learning rate with strong decay) to avoid overfitting and preserve CLIP’s generalization and text–image alignment, mitigating domain shift on novel categories.
>
> **W4**
>
> **Response:** We agree that comparing with alternative quality-aware heads would further strengthen the empirical analysis. We will add a detailed discussion comparing our LQEH with alternative quality estimators like GFL and Varifocal in the final version.
>
> Prior works (e.g., GFL and VarifocalNet) have shown that incorporating quality-aware signals into classification is generally beneficial. Our method follows a similar intuition but introduces a more effective way of leveraging quality information, which is validated by consistent improvements over the baseline in our experiments.
>
> Furthermore, we expect LQEH to be complementary to these existing heads, potentially yielding additional gains when combined. We will provide these comparisons and combination results in the revision to ensure a more comprehensive evaluation.
>
> **W5 and Q2**
>
> Response: We thank the reviewer for the comments. Our weaker teacher analysis follows the standard OVD protocol (P5, L245), using a DETR-style encoder trained strictly on COCO base classes to prevent novel-class leakage. Under this setup, our method consistently surpasses CLIPSelf by +6.1 AP (ViT-B/16) and +4.9 AP (ViT-L/14).
>
> To verify robustness, we conducted cross-dataset transfer (Table R3) and scale variation (Table R4) experiments. Our approach remains competitive on Objects365 and outperforms the baseline across all scales/novel categories on OV-COCO. These results demonstrate that our method generalizes effectively without being significantly affected by base-class supervision biases.
>
> **W6**
>
> **Response:** thank you for this helpful suggestion. We will include a more comprehensive diagnostic analysis in the final version, with representative visual examples and discussion of typical failure modes on novel categories, to better guide future improvements and real-world deployment.
>
> **W7**
>
> **Response:** Existing 3D OVD methods (e.g., [1, 2]) typically share a design paradigm with 2D OVD, where localization and classification are decoupled, leveraging CLIP-like models for open-vocabulary category prediction. Our work focuses on mitigating global semantic bias and enhancing spatial awareness via spatial distillation—contributions that are largely orthogonal to the dimensionality of the detection pipeline.
>
> Consequently, our framework can be naturally extended to 3D OVD by replacing the 2D teacher with 3D-aware pretrained models (e.g., SAM-3D or Point-BERT) to facilitate geometric and spatial representation learning. Since our method operates at the feature level and is independent of category-specific supervision, it can be seamlessly integrated into existing 3D OVD frameworks. We will include this discussion in the revised manuscript.

---

> > ### Author Rebuttal · Reviewer_oYQS · 2026-04-03
> >
> > Thanks the authors for the responses in detail. My concerns are addressed with evidences, like the computational overhead and dependence on base-supervised components. I tend to accept this submission. I will raise my score to 5.

---

### Official Review · Reviewer_Ckvv · 2026-03-11

**Soundness:** 4
**Presentation:** 4
**Significance:** 4
**Originality:** 1
**Overall Recommendation:** 4
**Confidence:** 5

**Summary:**

This paper proposes Object-level Semantic and Spatial Distillation (OSSD) to decouple the learning processes of semantic and spatial features. The model first distills object-level semantic features from the CLIP model and then distills spatial features from a pre-trained base-category detector. In addition, the paper introduces a Localization Quality Estimation Head (LQEH) module. The proposed method achieves significant performance improvements on the OV-COCO and OV-LVIS benchmarks.

**Compliance With Llm Reviewing Policy:**

Affirmed.

**Final Justification:**

My concerns have been fully addressed, so I will keep my positive evaluation.

**Key Questions For Authors:**

Please provide a more detailed analysis of the computational overhead. For reference, this could include the FLOPs, training time, and memory consumption introduced by the multiple teacher models and the two-stage distillation process.

**Limitations:**

yes

**Strengths And Weaknesses:**

Strengths:
1、The method achieves significant performance improvements, outperforming many existing baseline approaches.
2、The authors provide in-depth analyses of the proposed modules. Table 3 (ablation study), Table 4 (attention sensitivity analysis), and Table 5 (IoU supervision function) help explain the specific contribution of each design choice to the final performance.

Weaknesses:
1、 The proposed framework appears quite cumbersome. It involves multiple stages: training an RPN, semantic distillation using a frozen CLIP teacher, spatial distillation using another pre-trained detector as a teacher, and an S2S adapter.
2、Most of the baseline models used for comparison are from 2023 or early 2024. The authors should consider comparing their method with more recent and representative research works.
3、The concept of decoupling localization and classification, as well as multi-stage distillation, is relatively conventional in the OVD community. While the specific combination here is effective, the conceptual leap from existing distillation-based methods feels somewhat limited.

---

> ### Author Rebuttal · Authors · 2026-03-30
>
> We thank the reviewer for the valuable comments and insightful suggestions. We address each concern below.
>
> **W1**
>
> **Response:** Thank you for the insightful comment. While the framework appears complex, the multi-stage design is necessary due to the difficulty of jointly optimizing semantic alignment and spatial localization with CLIP features.
>
> We attempted a unified distillation objective, but found that under a frozen text encoder, CLIP cannot simultaneously achieve strong semantic alignment and spatial awareness, leading to suboptimal performance due to conflicting objectives.
>
> Therefore, we decouple the process into two stages: semantic distillation for aligning with text embeddings, and spatial distillation via an S2S adapter to enhance localization. As shown in Fig. 1(b), patch-level features already exhibit partial alignment, reducing the burden of semantic learning.
>
> Despite the additional components, the overall training cost is not increased. As shown in Fig. 6, our method converges significantly faster, reaching optimal performance within 10 epochs (~2× faster than the baseline), which compensates for the added complexity.
>
> **W2**
>
> **Response:** We thank the reviewer for this important comment. We agree that including more recent works can further strengthen the evaluation. To address this concern, we include comparisons with recent representative methods [1–2]. As shown in Table R1, R2, our method consistently outperforms them on OV-COCO and OV-LVIS.We will incorporate these results and expand the discussion in the final version.
>
> **Table R1: Results on OV-COCO benchmark.**
>
> | Method | Pre-train Model | Require Novel Class | $AP_{50}^{Novel}$ |
> | :--- | :---: | :---: | :---: |
> | CoT-PL[1] | RN50 | ✓ | 43.4 |
> | CoT-PL[1]  | RN50x4 | ✓ | 47.8 |
> | VMCNet[2] | VMCNet-L | ✗ | 48.5 |
> | **OSSD+DETR Encoder** | ViT-B/16 | ✗ | 43.7 |
> | **OSSD+DINOv2** | ViT-B/16 | ✗ | 44.9 |
> | **OSSD+DETR Encoder** | ViT-L/14 | ✗ | 49.2 |
> | **OSSD+DINOv2** | ViT-L/14 | ✗ | 52.4|
>
> **Table R2: Results on OV-LVIS benchmark.**
>
> | Method | Pre-train Model | Require Novel Class | $mAP_{r}$ |
> | :--- | :---: | :---: | :---: |
> | CoT-PL[1] | RN50x4 | ✓ | 26.4 |
> | VMCNet[2] | VMCNet-L | ✗ | 38.4 |
> | **OSSD+DETR Encoder** | ViT-L/14 | ✗ | 40.5 |
> | **OSSD+DINOv2** | ViT-L/14 | ✗ | 41.8 |
>
> [1] Choi, H., Lim, Y., Shin, J., et al. Cot-pl: Visual chain-of-thought reasoning meets pseudo-labeling for open-vocabulary object detection. arXiv preprint arXiv:2510.14792, 2025.
>
> [2] Gao, X., Dai, Y., Qiu, B., et al. Modulating cnn features with pre-trained vit representations for open-vocabulary object detection. arXiv preprint arXiv:2501.16981, 2025.
>
> **W3**
>
> **Response:** We thank the reviewer for this insightful comment. While decoupling and distillation are established in OVD, our contribution lies in explicitly decoupling knowledge sources and feature representations, rather than tasks.
>
> Conventional distillation-based OVD methods align a single unified feature map with CLIP, creating an inherent conflict: CLIP is optimized for image-level semantics but lacks spatial fidelity, leading to a trade-off between semantic consistency and localization accuracy.
>
> Our method (OSSD) addresses this by shifting from “task-level decoupling” to “source-level decoupling.” Specifically, we use CLIP solely for global semantic supervision, while a base-trained detector teaches spatial and structural knowledge, avoiding incompatible objectives in a shared representation.
>
> Overall, by alleviating CLIP’s spatial limitations through source-decoupled distillation, our method represents a meaningful conceptual advance, which we will further clarify in the final version.
>
> **Q1**
>
> **Response:** We thank the reviewer for the insightful suggestion. We provide a detailed analysis of the computational overhead of the multi-teacher framework and two-stage distillation, including FLOPs, training time, and memory (Tables R1–R2).
>
> Compared to the baseline, our full model (OSSD) incurs only a modest increase in inference cost (6043.28G, +8.3%). As shown in Table R2, training time is reduced from 34h to 20h due to faster convergence (Fig. 6), while the two distillation stages are lightweight, requiring only 9h and 2h.
>
> Overall, although multi-teacher distillation adds some training overhead, it is well controlled without compromising efficiency, and inference cost remains nearly unchanged, ensuring practical deployment.
>
> **Table R1: Model Complexity**
>
> |Method|Input|FLOPs|Params|Trainable|
> |---|---|---|---|---|
> |Baseline|(896,896)|5579.42G|451.33M|23.17M|
> |OSSD|(896,896)|6043.28G|460.77M|23.25M|
> |Sem. distill|(896,896)|1190.82G|877.43M|315.05M|
> |Spa. distill|(896,896)|862.89G|741.89M|9.44M|
>
> **Table R2: Training Efficiency**
>
> |Method|BS|Speed(s/b)|Time|Mem|
> |---|---|---|---|---|
> |Baseline|4|3.787|34h|21G|
> |OSSD|4|2.859|20h|20G|
> |Sem. distill|1|0.540|9h|18G|
> |Spa. distill|32|2.490|2h|18G|

---

> > ### Author Rebuttal · Reviewer_Ckvv · 2026-04-04
> >
> > The author has fully addressed my concerns.

---

### Official Review · Reviewer_SDPd · 2026-03-12

**Soundness:** 3
**Presentation:** 3
**Significance:** 3
**Originality:** 3
**Overall Recommendation:** 4
**Confidence:** 3

**Summary:**

This paper proposes a distillation learning model framework for Open-Vocabulary Object Detection, utilizing both semantic and spatial distillation methods. The proposed head design based on Hungarian matching, along with the two distillation methods, significantly outperforms existing SOTA methods on datasets such as OV-COCO.

**Compliance With Llm Reviewing Policy:**

Affirmed.

**Final Justification:**

The authors addressed my questions well, and I maintain my original evaluation.

**Key Questions For Authors:**

1. In Table 3, it is suggested to add the experimental results for Semantic Distillation + LQEH.
2. In Table 2, it is suggested to include the results for pure OSSD (without using Spatial Distillation), as the current comparison may be unfair.
3. In Table 1, it would be beneficial to compare the ViT framework with recent works.

**Limitations:**

yes

**Strengths And Weaknesses:**

Soundness: The paper is relatively reliable, with rich arguments and sufficient details. The theoretical explanations are fairly clear.

Presentation: The presentation is good, with a simple and clear method structure, accurate formulas, and clear, mobile figures. The overall model is easy to reproduce.

Significance: The significance is moderate. The main issue lies in the fairness of the comparisons. The paper uses three modules, with the Spatial Distillation part utilizing the additional DINOv2 model. Although the Semantic Distillation outperforms the 2023 CLIPSelf method with the same backbone, it is not significantly outstanding.

Originality: The originality is high. Although the problem insights are similar to previous methods, different solutions are adopted, and the three sub-modules have certain innovations.

---

> ### Author Rebuttal · Authors · 2026-03-30
>
> We thank the reviewer for the valuable comments and insightful suggestions. We address each concern below.
>
> **W1 and Q2.**
>
> **Response:** Thank you for the valuable comments. We clarify that our framework includes two variants of spatial distillation:
> (a) a DETR-style encoder trained on COCO base classes, following the standard OVD protocol (Page 5, Line 245);
> (b) a DINOv2-based teacher, used to study the scalability of our method with stronger visual representations (Page 7, Line 376).
>
> Importantly, our primary comparisons are conducted under the standard OVD setting (COCO-trained DETR encoder), ensuring no additional novel-class knowledge is introduced. The DINOv2 results are reported as supplementary analysis to demonstrate robustness rather than for direct comparison.
>
> Regarding spatial distillation, its role is not to transfer semantic knowledge, but to enhance spatial awareness (e.g., object boundaries and localization sensitivity), which CLIP inherently lacks. This improves its suitability for detection without introducing extra supervision.
>
> Under the standard open-vocabulary detection setting---where a DETR-style encoder is trained on COCO base classes only---our method achieves consistent gains over CLIPSelf ($+$6.1 AP on ViT-B/16 and $+$4.9 AP on ViT-L/14), demonstrating the effectiveness of our design.
>
>
> **Q1**
>
> **Response:** Thank you for your helpful suggestion. We have expanded Table 3 to conduct experiments combining Semantic Distillation + LQEH (Table 3, Experiment E). We observed that this combination leads to a performance drop of 4.7 AP compared to using Semantic Distillation alone. A preliminary analysis suggests that LQEH relies on high-quality spatial representations for effective training. When only Semantic Distillation is applied (without Spatial Distillation), the CLIP backbone may lack sufficiently strong spatial features, which could limit the effectiveness of LQEH and result in suboptimal performance.
>
> In future work, we plan to conduct more controlled experiments to better understand this interaction, such as replacing LQEH with alternative class-agnostic detection heads to validate this hypothesis.
>
> **Table 3: Ablation Results**
>
> |          | Spatial Distillation | Semantic Distillation | LQEH | $AP_{50}^{Novel}$ |
> | -------- | -------- | -------- | -------- | -------- |
> | baseline | -                    | -                     | -    | 38.3        |
> | A        | ✓                    | ✗                     | ✗    | 42.3        |
> | B        | ✗                    | ✓                     | ✗    | 45.0        |
> | C        | ✗                    | ✗                     | ✓    | 41.7        |
> | D        | ✓                    | ✓                     | ✗    | 47.8        |
> | E        | ✗                    | ✓                     | ✓    | 40.3        |
> | F        | ✓                    | ✓                     | ✓    | 49.2        |
>
> **Q3**
>
> **Response:** We appreciate this constructive feedback. We have included comparisons with recent methods leveraging MLLMs and other pretrained models [1-2]. Notably, our method still maintains superior performance on both OV-COCO and OV-LVIS benchmarks, as shown in Table R1 and R2. We attribute this to our OSSD framework and class-agnostic quality estimator, which provide more reliable novel object perception without over-relying on complex external priors. We will incorporate this comparison and corresponding literature review to the final version of the paper.
>
> **Table R1: Results on OV-COCO benchmark.**
>
> | Method | Pre-train Model | Require Novel Class | $AP_{50}^{Novel}$ |
> | :--- | :---: | :---: | :---: |
> | CoT-PL[1] | RN50 | ✓ | 43.4 |
> | CoT-PL[1] | RN50x4 | ✓ | 47.8 |
> | VMCNet[2] | VMCNet-L | ✗ | 48.5 |
> | **OSSD+DETR Encoder** | ViT-B/16 | ✗ | 43.7 |
> | **OSSD+DINOv2** | ViT-B/16 | ✗ | 44.9 |
> | **OSSD+DETR Encoder** | ViT-L/14 | ✗ | 49.2 |
> | **OSSD+DINOv2** | ViT-L/14 | ✗ | 52.4|
>
> **Table R2: Results on OV-LVIS benchmark.**
>
> | Method | Pre-train Model | Require Novel Class | $mAP_{r}$ |
> | :--- | :---: | :---: | :---: |
> | CoT-PL[1] | RN50x4 | ✓ | 26.4 |
> | VMCNet[2] | VMCNet-L | ✗ | 38.4 |
> | **OSSD+DETR Encoder** | ViT-L/14 | ✗ | 40.5 |
> | **OSSD+DINOv2** | ViT-L/14 | ✗ | 41.8 |
>
> [1] Choi, H., Lim, Y., Shin, J., et al. Cot-pl: Visual chain-of-thought reasoning meets pseudo-labeling for open-vocabulary object detection. arXiv preprint arXiv:2510.14792, 2025.
>
> [2] Gao, X., Dai, Y., Qiu, B., et al. Modulating cnn features with pre-trained vit representations for open-vocabulary object detection. arXiv preprint arXiv:2501.16981, 2025.

---

> > ### Author Rebuttal · Reviewer_SDPd · 2026-04-03
> >
> > The author's response effectively addresses my question. The interaction between semantic distillation and LQEH in Q1 can be added to the main text to guide the reader in understanding the relationships and limitations of the model's various modules.

---

### Decision · Program_Chairs · 2026-04-30

**Decision:**

Accept (regular)

**Comment:**

This paper received overall positive reviews. All reviewers initially recommended weak accept. They generally regard the proposed framework novel and solid, the writing clear, the improvement over existing baselines strong, and the ablation study convincing. There are also some concerns about the framework design, outdated baselines, computational overhead, etc.  The rebuttal was persuasive. After the rebuttal, Reviewer oYQS further increased the rating from weak accept to accept. The AC checked the paper, rebuttal, and review comments, and recommends accepting the paper.